# An Integrative Simulation for Mixing Different Polycarbonate Grades with the Same Color: Experimental Analysis and Evaluations

**Jamal Alsadi *** , **Rabah Ismail and Issam Trrad**

Engineering College, University of Jadara, Irbid 21110, Jordan; r.ismail@jadara.edu.jo (R.I.); i.trrad@jadara.edu.jo (I.T.)
* Correspondence: jamal.alsadi@ontariotechu.net

**Abstract:** The processing parameters' impact such as temperature (Temp.), feed rate (F.R.), and speed (S.) at three distinct grades of the same color was explored in this study. To investigate the effect of the characteristics on color formulations, they were each adjusted to five different levels. For these grades, which were all associated with the same color, an intermeshing twin-screw extruder (TSE) was used. The compounded materials were molded into flat coupons then evaluated with a spectrophotometer for their CIE (L*, a*, b*, and dE*) values. A spectrophotometer was used to determine the color of a compounded plastic batch, which measured three numbers indicating the tristimulus values (CIE L*a*b*). The lightness axis, which ranged from 0 (black) to 100 (white), is known as the L*-axis (white). Redness-greenness and yellowness-blueness were represented by the other two coordinates, a* and b*, respectively. The color difference deviation (Delta E*) from a target was dimensionless, when dE* approached zero. However, the most excellent favorable color difference value occurred and different processing impact factors on polycarbonate grade were investigated. Using the response service design (RSD) software of Stat-Ease Design-Expert® (Minneapolis, MN, USA), historical data were gathered and evaluated. To reduce the value of dE*, the impacts of these processing factors were investigated with the three processing parameters. The whole tristimulus color value could be simulated. Parameters were adjusted on 45 different treatments, using a five-level controlled response method to investigate their impact on color and detect non-optimal responses. The ANOVA for each grade was used to build the predicted regression models. The significant processing parameters were subjected to experimental running to simulate the regression models and achieve the best color, reducing waste.

**Keywords:** different grades; RSD; simulate r egression models; processing and parameters; analysis of variance; resin pigment blends



## 1. Introduction

Plastics are relatively new materials for producing colored materials. As a result, there are few scientific data on plastic color mismatching and its long-term consequences. Plastic fabrication allows for the creation of robust, lightweight plastics in various shapes. In many situations, plastic shapes are favored over metal shapes. Polycarbonate is a rigid, transparent polymer used in a variety of applications. Some factors may significantly impact the color of plastics intended for outdoor use. As a result, it is critical to comprehend how numerous elements can influence material compounding. This research aims to see how processing settings affect color matching for a few grade-color. To generate the proper color with minimal waste, the plastic industry has spent the last few decades seeking to understand the significant challenges involved in plastic color matching procedures. Lambert's law claims that the amount of light absorbed is proportional to the concentration of the absorbing substance. Still, Beers law states that the amount of light absorbed is proportional to the thickness of the absorbing material [1]. Manufacturing technology

makes colored plastic for plastic process prototyping on a small and medium scale. As a result, the plant receives orders that must be completed in a matter of days.

To summarize, an object's color appearance is determined by the total amount and type of scattering and absorption that occurs. As a result, the item will seem white if there is no absorption and nearly equal levels of scattering at all visible wavelengths; and if the visible light is absorbed by the pigment [2].

Several hundred ingredients are divided into three categories: resins, additives, and pigments. Ingredients and additives combine to create a specific grade of plastic. The pigments give the plastic its color. Because of their surfaces and orientations, the pigments absorb certain hues, while reflecting others randomly.

White light is created by mixing all visible spectrum wavelengths in roughly equal quantities [3]. The light source and observer are replaced with color measurement tools such as a colorimeter or spectrophotometer to standardize color evaluation in the plastic compounding industry [4].

As color is represented in codes or values, this allows for more uniform color recognition. There were two data mining approaches used. One was a decision tree classifier, and the other was online analytical processing (OLAP) (DTC). OLAP assisted in identifying a relationship between factors that resulted in failed batches and parameters with a high rate of alteration. The DTC was proposed as a decision assistance tool for detecting combinations of characteristics that could cause a color mismatch. The DTC investigates characteristics that could lead to color mismatch issues in compounded polymers. To find such factors in the past, OLAP and data mining methodologies were applied [5–7].

Overall, DTC was utilized in this study to investigate possible correlations between the components of color, grade, kind, product, and line. To the best of our knowledge, no research has used DTC for color mismatch analysis, according to our literature review. DTC is used in some relevant manufacturing articles (semi-conductors [8]). Other researchers have used neuro-networks to forecast output colors based on past data [9].

In previous studies, the artificial neural network (ANN) was utilized to eliminate mistakes in polycarbonate color values [10]. The neural network in this paper was used to reduce the errors in color tristimulus values (L*, a*, b*), which directly affect the D.E. calculated [11].

The problem cannot be solved by concentrating on a few situations because the colors' nature is constantly changing. Therefore, this research proposal focuses on determining the fundamental causes of color mismatches in compounded plastics. As a result, plastics firms will reduce waste and boost production. More importantly, it will improve knowledge of the technical challenges of color matching in plastic production. Focusing on resolving challenges for a single product is complex and potentially fruitless because that product may not be duplicated in the future. Compared to the paint industry, color mismatch issues have not been investigated deeply through the plastic compounding business. The parameter(s) creating first-pass color opportunities must be discovered to limit material rejects. Researchers mixed three different titanium dioxide pigments into heavily loaded polyethylene masterbatches, each with a different surface treatment.

They discovered that the three grades' best screw design and operating conditions were considerably different. Processing circumstances or certain combinations of modifiers and additives in the resin system were shown to have a negative impact on the final desired hue [12,13]. Paints and coatings have had a lot of research done on pigment dispersion, but plastics have not gotten nearly as much attention [14,15].

The high shear rates, processing temperatures, and processing pressures used in plastics manufacturing operations [16,17] significantly contrast the two dispersion mechanisms. Various scholars have conducted several investigations, during compounding, on the effect of processing parameters on color [18,19]. The minimum processing time is advised to achieve excellent gloss, brilliance, and blend uniformity. Furthermore, for each item, an optimal loading should be utilized; too much pigment is not only expensive but also hazardous because it diminishes impact resistance [20,21]. Increase the duration of the

mixing and decrease the viscosity of resin to solve problems with dispersion or achieving a homogeneous mixture [22,23]. Various researchers have conducted a few investigations on the influence of processing factors on dynamic mixing in a screw extrusion during polymer compounding [18,24].

Many scientists have reviewed polymer blending as an essential field of polymer science. As Sanchez et al. [25] demonstrated, the PC/PBT blends are transparent in the melt stage and somewhat miscible blends in the solid state.

Liang and Gupta (2000) investigated the rheological qualities of a recycled P.C. blended with virgin P.C., concluding that separated P.C. could be added to pure P.C. up to 15% without significantly affecting its properties [26]. Lee S. et al. investigated the rheological and phase behavior of P.C./Polyester blends. They discovered, however, that the combinations do not obey the mixing rule, which is standard in all investigations. They discovered, however, that the combinations do not obey the mixing rule, which is standard in all investigations [27]. Other researchers' experiments on extruders showed that single screw extruders could reach dispersive mixing capabilities comparable to twin-screw extruders [28].

A 45-mm diameter single-screw extruder with eight glass panes was used in another investigation to investigate the color mixing process [29]. The researchers determined where color mixing began and finished by using such an extruder. The quality of mixing was shown to be directly proportional to the maximum processing pressure in the extruder. Furthermore, earlier research has examined how the screw shape and operating conditions affect dispersion performance and torque loading during twin-screw compounding [30].

One of the most significant color matching components taken from a remote location is spectrophotometric measures to create a suitable color standard. Spectrophotometers are valuable quality control equipment for measuring color and defining color variations numerically. However, their function as a device is to reduce a color target to a collection of numbers, which are subsequently sent to a color formulator as a matching target [31,32]. CIELAB is the name of the color measurement method. The values utilized by CIE are named L*, a*, and b*. L* denotes the difference between light (L* = 100) and dark (L* = 0), a* denotes the green (−a*) and red (+a*) difference, and b* indicates the yellow (+b*) and blue (−b*) difference [33,34]. dE* is used to express deviations in L*, a*, and b*, where:

$$dE^* = \sqrt{(\Delta L^*)^2 + (\Delta a^*)^2 + (\Delta b^*)^2} \tag{1}$$

The color difference's amplitude, not its direction, is represented by dE*. As a quality control measure, colored materials are compared to a standard when being manufactured. Color discrepancies are employed instead of absolute color values. The total color change, dE*, shows the color difference in the CIELAB color space [17,35].

The findings of designed experiments were analyzed and discussed in this study, which highlights individual and combined influences on output color of three process parameters.

The experimental data confirm the statistical model's fitness [36] by systematically examining resins, additives, and pigments, and how processing conditions and diverse interactions impact them. More precisely, the scientific concerns surrounding the twin co-rotating screw process processing parameters on different grades of the same color were explored. The study's main aim was to develop an equation that might be used to determine differences between the two samples and could be used to any color at any time. To explore the impact of parameters on color and detect non-optimal responses, a five-level controlled response method was used on 45 different treatments. The anticipated regression models were built using the ANOVA for three different grades. Speed, temperature, and F.R. were among the processing characteristics studied. To provide a foundation for process improvement recommendations, experimental data were collected, and statistical analysis was undertaken.

## 2. Materials and Methods

The three classes with the highest adjustment when dealing with red pigments were discovered based on preliminary data mining results from the first few months of 2009. In this study, these grades were denoted by the numbers 1, 2, and 3. For the dispersion of color in parts per 100 among these grades, a mixture of two polycarbonate resins and four distinct pigments were utilized (PPH). As indicated in Table 1, all three grades utilized the same color, as were shown in Figure 1.

**Table 1.** Compounding formulation used for three grades.

| Resin/Color | Grade–Color (1) | | Grade–Color (2) | | Grade–Color (3) | |
|---|---|---|---|---|---|---|
| Type | pph | gms | pph | gms | pph | gms |
| Resin 1 | 30 | 1800 | 30 | 1800 | – | – |
| Resin 2 | 70 | 4200 | 70 | 4200 | 100 | 6000 |
| White Pigment | 1.925 | 115.5 | 1.76 | 105.6 | 1.76 | 105.6 |
| Black Pigment | 0.11 | 6.60 | 0.00968 | 0.5808 | 0.00968 | 0.58 |
| Red Pigment | 0.1875 | 11.25 | 0.01602 | 0.9612 | 0.01602 | 0.96 |
| Yellow Pigment | 0.1075 | 6.45 | 0.1084 | 6.504 | 0.1084 | 6.50 |

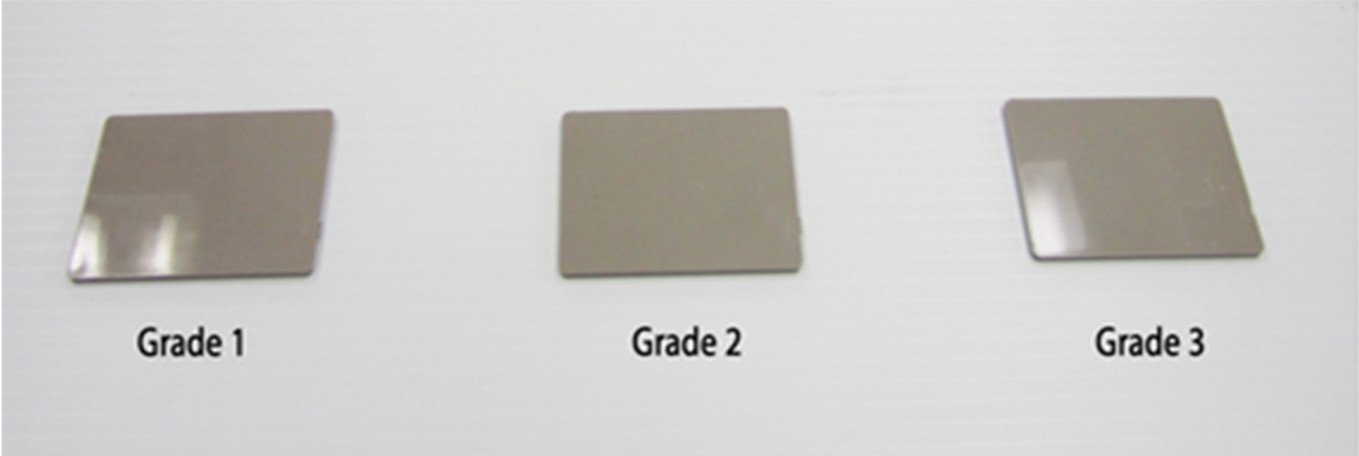

**Figure 1.** Three grades have different formulations but the same color.

Grades 1 and 2 used a mixture of two polycarbonates resins with various weights of the same pigments, while grade 3 used one poly car resin with the same weight of pigments as grade 2. As a result, resin 1 had a melt flow index (MFI) of 25 g/min, while resin 2 had an MFI of 6.5 g/10 min, where the weights were heavier than water, and the temperature for autoignition was 630 °C for all grades. At the industrial plant, three grades were subjected to testing. The materials were extruded at L/D ratios of 37 and Do/Di ratios of 1.55, respectively, utilizing a twin-screw extruder (25.5 mm, 27 kW). There were ten heating zones on the extruder, nine designated on the barrel, and one at the die.

The extruded melt was then pelletized after being quenched in cold water. These pellets were subsequently formed into rectangular chips (3 × 2 × 0.1 inches), which were measured against a target value via injection molding. Three coupons were created for each experiment at each of the five-parameter values to assure accuracy. Then each voucher was given three readings. The total simulating design data for the tristimulus color value with the three processing parameters were 45 runs, as recorded in Table 2.

**Table 2.** Response surface design 45 runs for 3 grades.

| Nos. | Temp | RPM | kg/h | Grade | L* | a* | b* | dE* |
|------|------|-----|------|-------|-----|-----|-----|------|
| 1 | 230 | 750 | 25 | Grade 1 | 67.26 | 1.52 | 4.545 | 0.435 |
| 2 | 240 | 750 | 25 | Grade 1 | 67.1767 | 1.5 | 4.5 | 0.456 |
| 3 | 255 | 750 | 25 | Grade 1 | 67.285 | 1.43 | 4.453 | 0.533 |
| 4 | 270 | 750 | 25 | Grade 1 | 67.185 | 1.511 | 4.56167 | 0.4 |
| 5 | 280 | 750 | 25 | Grade 1 | 66.735 | 1.547 | 4.63 | 0.5 |
| 6 | 255 | 700 | 25 | Grade 1 | 67.055 | 1.48167 | 4.41167 | 0.55 |
| 7 | 255 | 725 | 25 | Grade 1 | 67.0333 | 1.46667 | 4.34667 | 0.62 |
| 8 | 255 | 750 | 25 | Grade 1 | 67.286 | 1.49 | 4.45 | 0.54 |
| 9 | 255 | 775 | 25 | Grade 1 | 66.995 | 1.44167 | 4.30167 | 0.66 |
| 10 | 255 | 800 | 25 | Grade 1 | 67.1033 | 1.45167 | 4.30833 | 0.65 |
| 11 | 255 | 750 | 20 | Grade 1 | 67.0183 | 1.55 | 4.78 | 0.22 |
| 12 | 255 | 750 | 23 | Grade 1 | 66.81 | 1.423 | 4.41 | 0.63 |
| 13 | 255 | 750 | 25 | Grade 1 | 67.285 | 1.5 | 4.45 | 0.54 |
| 14 | 255 | 750 | 27 | Grade 1 | 66.7583 | 1.43 | 4.41 | 0.65 |
| 15 | 255 | 750 | 30 | Grade 1 | 66.915 | 1.43 | 4.47 | 0.53 |
| 16 | 230 | 750 | 25 | Grade 2 | 66.44 | 1.57 | 4.71 | 1.29 |
| 17 | 240 | 750 | 25 | Grade 2 | 66.33 | 1.54 | 4.63 | 1.28 |
| 18 | 255 | 750 | 25 | Grade 2 | 66.37 | 1.56 | 4.77 | 1.25 |
| 19 | 270 | 750 | 25 | Grade 2 | 66.47 | 1.54 | 4.65 | 1.24 |
| 20 | 280 | 750 | 25 | Grade 2 | 66.21 | 1.55 | 4.68 | 1.23 |
| 21 | 255 | 700 | 25 | Grade 2 | 66.3533 | 1.55167 | 4.74167 | 1.228 |
| 22 | 255 | 725 | 25 | Grade 2 | 66.4183 | 1.54833 | 4.73 | 1.16 |
| 23 | 255 | 750 | 25 | Grade 2 | 66.3733 | 1.56 | 4.77667 | 1.21 |
| 24 | 255 | 775 | 25 | Grade 2 | 66.3017 | 1.57167 | 4.80833 | 1.278 |
| 25 | 255 | 800 | 25 | Grade 2 | 66.5067 | 1.55667 | 4.76 | 1.21 |
| 26 | 255 | 750 | 20 | Grade 2 | 66.575 | 1.56667 | 4.67167 | 1.018 |
| 27 | 255 | 750 | 23 | Grade 2 | 66.465 | 1.582 | 4.676 | 1.128 |
| 28 | 255 | 750 | 25 | Grade 2 | 66.3733 | 1.56 | 4.77667 | 1.21 |
| 29 | 255 | 750 | 27 | Grade 2 | 66.345 | 1.585 | 4.71 | 1.24 |
| 30 | 255 | 750 | 30 | Grade 2 | 66.4783 | 1.58667 | 4.69333 | 1.11 |
| 31 | 230 | 750 | 25 | Grade 3 | 67.715 | 1.63 | 5.115 | 0.4 |
| 32 | 240 | 750 | 25 | Grade 3 | 67.515 | 1.686667 | 5.236667 | 0.51 |
| 33 | 255 | 750 | 25 | Grade 3 | 67.515 | 1.686667 | 5.236667 | 0.46 |
| 34 | 270 | 750 | 25 | Grade 3 | 67.605 | 1.641667 | 5.113333 | 0.38 |
| 35 | 280 | 750 | 25 | Grade 3 | 67.525 | 1.68 | 5.235 | 0.51 |
| 36 | 255 | 700 | 25 | Grade 3 | 67.48 | 1.66 | 5.18 | 0.438 |
| 37 | 255 | 725 | 25 | Grade 3 | 67.5583 | 1.615 | 5.10167 | 0.41 |
| 38 | 255 | 750 | 25 | Grade 3 | 67.515 | 1.68667 | 5.23667 | 0.46 |
| 39 | 255 | 775 | 25 | Grade 3 | 67.8467 | 1.60833 | 5.03833 | 0.42 |
| 40 | 255 | 800 | 25 | Grade 3 | 67.525 | 1.68 | 5.235 | 0.39 |
| 41 | 255 | 750 | 20 | Grade 3 | 67.6283 | 1.63833 | 5.065 | 0.346 |
| 42 | 255 | 750 | 23 | Grade 3 | 67.5733 | 1.63833 | 5.10667 | 0.378 |
| 43 | 255 | 750 | 25 | Grade 3 | 67.515 | 1.68667 | 5.23667 | 0.463 |
| 44 | 255 | 750 | 27 | Grade 3 | 67.635 | 1.64167 | 5.13 | 0.413 |
| 45 | 255 | 750 | 30 | Grade 3 | 67.54 | 1.63 | 5.12 | 0.38 |

In CIE L*, a*, and b* values, L* = 67.57, a* = 1.43, and b* = 4.8 were chosen as the required color output, while the permitted dE* was 0.85. Using the Software of Design-Expert, Version 8, Stat-Ease Inc. (Minneapolis, MN, USA), the statistical data were established. Then the data were used to compare and analyze the factors' effect on grades. The ANOVA determined which parameters were significant and whether there was any interaction between them. As previously stated, the study's goal: develop an equation that could help in expecting the L*, a*, and b* tristimulus values.

## 3. Results

The design of the experiment was used to do statistical analysis and ANOVA. Using Stat-Ease Design Expert® Version 8 software, the influence of parameters on L*, a*, b*, and dE* was investigated, as seen in Table 3.

**Table 3.** Summary of the design data.

| Type | Response Surface | Run | 45 | Response Surface Design |
|---|---|---|---|---|
| Design Type | Historical Data | Blocks | No Blocks | |
| Design Model | Quadratic | Build time | 59.3 | |
| Factor | Name | Units | Type | Sub-Type |
| A | Temp | °C | Numeric | continuous |
| B | Speed | RPM | Numeric | continuous |
| C | Feed rate | kg/h | Numeric | continuous |
| D | grade | | Categoric | Nominal |
| Factor | Min | max | Coded | Values |
| A | 230 | 280 | −1 | 1 |
| B | 700 | 800 | −1 | 1 |
| C | 20 | 30 | −1 | 1 |
| D | B | A | | |
| RESPONSE | Name | Obs | Analysis | Model |
| Y1 | L* | 37 | Polynomial | R Linear |
| Y2 | a* | 37 | Polynomial | Quadratic |
| Y3 | b* | 37 | Polynomial | $R^2$ Fi |
| Y4 | dE* | 37 | Polynomial | Quadratic |
| RESPONSE | Min | Max | Mean | Std.Dev |
| Y1 | 66.21 | 68 | 67.02 | 0.5 |
| Y2 | 1.43 | 1.7 | 1.56 | 0.07 |
| Y3 | 4.3 | 5.2 | 4.7 | 0.299 |
| Y4 | 0.22 | 1.3 | 0.7 | 0.36 |

Note: A Temp, B Speed, C Feed Rate, D grade, Y1 L*, Y2 a*, Y3 b* and Y4 dE*.

### 3.1. Analysis of Variance (ANOVA)

Sequential F-tests were run using a linear model as a starting point and adding terms (quadratic and linear if appropriate). The F-statistic was assessed for each model type, and the highest degree and critical elements model was picked. The same procedure was used for all tristimulus values, and only the significant terms were included. The ANOVA table for the sum of squares of a sequential model for dE* characterization is shown in Table 4. The quadratic model with the Prob > F was < 0.05, the most considerable condition. Furthermore, it was statistically significant (Prob > F was less than 0.0001) because it had a high F value (184.4). As a result, this model was suitable for the dE* response. The model's adjusted R-square value (97%) also corroborated this, as seen in Table 4.

The adjusted R-square measure was the same as the R-square measure, except that it was scaled down to account for the number of variables in the model. Both measures represent the model's capacity to explain variation in the answer. For example, the observed adjusted R-square value of 97% showed that the model explained roughly 97% of the variability in dE*. In contrast, about 3% of the variability in dE* was unknown.

The adjusted R-square value of 0.96 was reasonably close to the predicted R-square value of 0.95. A signal-to-noise ratio was used in the Adeq Precision measurement. It is ideal to have a ratio of more than four. The observed percentage of 35.5 specified that the observed variance is significant compared to the fitted model's underlying uncertainty. In other words, the observed variance was significant in proportion to the fitted model's underlying uncertainty. The design space systems can also be generated by using the exact modeling.

**Table 4.** ANOVA for the color of three grades.

| Tristimulus Values | Processing Factors | F-Statistic Value | Probability Values | $R^2$ | Adjacent $R^2$ | Predicted $R^2$ | Adequate Precision |
|---|---|---|---|---|---|---|---|
| L* | Model | 193.82 | 0.0001 | 0.9463 | 0.9414 | 0.9318 | 34.082 |
| | A | 5.21 | 0.029 | | | | |
| | D | 288.12 | 0.0001 | | | | |
| a* | Model | 37.71 | 0.0001 | 0.901 | 0.8771 | 0.8385 | 20.869 |
| | A | 0.24 | 0.626 | | | | |
| | C | 3.46 | 0.073 | | | | |
| | D | 122.62 | 0.0001 | | | | |
| | CD | 5.2 | 0.0117 | | | | |
| | $A^2$ | 3.32 | 0.079 | | | | |
| b* | Model | 75.96 | 0.0001 | 0.9245 | 0.9124 | 0.8553 | 23.831 |
| | C | 1.05 | 0.3126 | | | | |
| | D | 185.69 | 0.0001 | | | | |
| | CD | 2.98 | 0.0654 | | | | |
| dE* | Model | 184.47 | 0.0001 | 0.9736 | 0.9683 | 0.9532 | 35.528 |
| | C | 10.63 | 0.0028 | | | | |
| | D | 538.19 | 0.0001 | | | | |
| | CD | 3.13 | 0.0583 | | | | |
| | $C^2$ | 21.44 | 0.0001 | | | | |

Feed rate (C) and grade (D) had significant effects on dE*, as shown in Table 4. Their *p*-values (Prob > F) were equal or less than 0.05 (typically ≤0.05), indicating that they were statistically significant models. On the other hand, temperature (A) and speed (B) had large *p*-values, indicating that they were not statistically significant for the dE* response. The interaction between feed rate (C) and the investigated grades (D) would be statistically significant if a confidence level of 90% was assumed; the *p*-value for this interaction (CD) in the model fitted to dE* was 0.0583 (see Table 4).

### 3.2. Simulate Regression Models

The expected response for each response was determined using multiple linear regression analysis. Equations (2)–(13) depict the response functions for grade 1, 2, and 3 for L*, a*, b*, and dE*, respectively, as shown in Table 5.

**Table 5.** Simulate regression models.

| Response | Regression Model | | |
|---|---|---|---|
| | Grade 1 | Grade 2 | Grade 3 |
| L* | $68.04585 - 3.931478 \times 10^3 \times \text{Temp} \ldots$ $\ldots \ldots \ldots \ldots \ldots \ldots \ (2)$ | $67.41030 - 3.93147 \times 10^3 \times \text{Temp} \ldots$ $\ldots \ldots \ldots \ldots \ (6)$ | $68.59188 - 3.93147 \times 10^3 \times \text{Temp} \ldots$ $\ldots \ldots \ldots \ldots \ldots \ldots \ldots \ (10)$ |
| a* | $3.97525 - 0.017160 \times \text{Temp} - 0.013080 \times \text{Feed Rate} + 3.399968 \times 10^5 \times \text{Temp2}$ $\ldots \ldots \ldots \ldots .. \ldots \ (3)$ | $3.67696 - 0.017160 \times \text{Temp} + 1.82759 \times 10^3 \times \text{Feed Rate} + 3.39996 \times 10^5 \times \text{Temp2} \ldots \ldots \ldots \ldots \ (7)$ | $3.82467 - 0.017160 \times \text{Temp} - 6.02931 \times 10^4 \times \text{Feed rate} + 3.39996 \times 10^3 \times \text{Temp2} \ldots \ (11)$ |
| b* | $5.26525 - 0.031351 \times$ $\text{Feed Rate} \ldots \ldots \ldots \ldots \ (4)$ | $4.62909 + 3.03966 \times 10^3 \times \text{Feed Rate}. \ldots$ $\ldots \ldots \ldots \ldots \ldots \ (8)$ | $5.00853 + 5.54586 \times 10^3 \times \text{Feed Rate} \ldots$ $\ldots \ldots \ldots \ldots \ldots \ldots \ (12)$ |
| dE* | $-3.68725 + 0.30417 \times$ $\text{Feed Rate} - 5.409068 \times 10^3 \times \text{Feed Rate2} \ldots \ldots \ldots \ldots \ (5)$ | $-2.44834 + 0.28225 \times \text{Feed Rate} - 5.409068 \times 10^3 \times \text{Feed Rate2} \ldots \ldots \ldots$ $\ldots \ldots \ (9)$ | $-3.04190 + 0.27459 \times \text{Feed Rate} - 5.409068 \times 10^3 \times \text{Feed Rate2} \ldots \ldots \ldots$ $\ldots \ldots \ldots \ldots \ldots \ (13)$ |

### 3.3. Point Prediction

The response surface method was optimized using a "numerical optimizer" for the lowest color value (dE*) in the feasible region. The Design-Expert response®'s (Minneapolis, MN, USA) optimizer calculated numerous local (feasible area) variables. For each grade, Table 6 provides the predicted tristimulus color values of CIE (L*, a*, b*, and dE*). Minor deviations were detected in the color values acquired by the optimization process. These discrepancies could be due to a lack of precise temperature control during the extrusion

process, which affects the viscosity of the polymer, as well as the pigment dispersion and ability to obtain the required color.

**Table 6.** Simulate tristimulus color solutions.

| Grade | Process. Parameters | | | Tristimulus Values | | | |
|---|---|---|---|---|---|---|---|
| | Temp | Screw Speed | Feed Rate | L* | a* | b* | dE* |
| | °C | rpm | kg/h | Black/White | Red/Green | Yellow/Blue | Color O.P. |
| 1 | 250.9 | 750 | 25.16 | 67.15 | 1.48 | 4.47 | 0.54 |
| 2 | 243.56 | 750 | 21.21 | 66.9 | 1.55 | 4.69 | 1.1 |
| 3 | 257.34 | 750 | 24.38 | 67.58 | 1.64 | 5.14 | 0.43 |

### 3.4. Effect of Processing Parameters through 3 Grades

Figures 2–11 show the impact of process conditions on color output over all three grades in terms of CIE dE* values, as created by Design-Expert® L*, a*, and b* represents CIE tristimulus data, related graphics were also created. However, because dE* takes precedence in this paper, the figures for these tristimulus values are limited. A common occurrence for the grades responsible for the reddest pigment modifications, a design of the experiment was carried out. The goal of the tests was to figure out what processing and material characteristics were producing color discrepancies. The color difference and the processing factors were explored for correlations. With Design-Expert®, general trends were charted, and trials were carried out and statistically analyzed. The experiment was designed with temperature, speed, and flow rate are three processing conditions for a total of 45 runs; these runs were done in three grades (1, 2, and 3) for one parameter while keeping the other two constants, as shown in Table 2.

Five levels were employed to conduct experiments on these three parameters. For example, temperatures reached 230, 240, 255, 270, and 280 °C. For each grade and level, values for the four responses (L*, a*, b*, and dE*) were taken from three different coupons and three different positions on each of these coupons. Figures 1–9 show the variation in color output for grades 1, 2, and 3 as a function of the three processing settings.

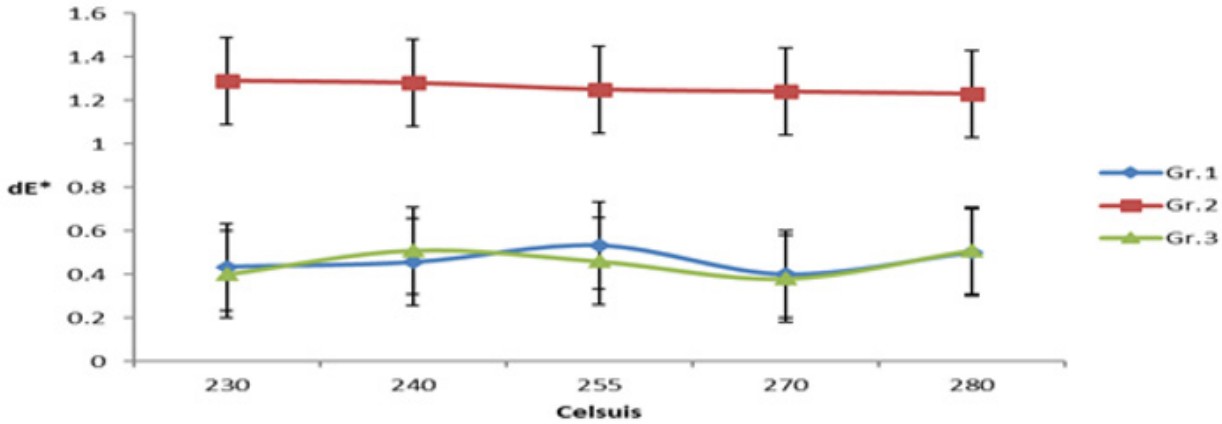

**Figure 2.** Temperature effects on dE* for Grade 1, Grade 2 and Grade 3.

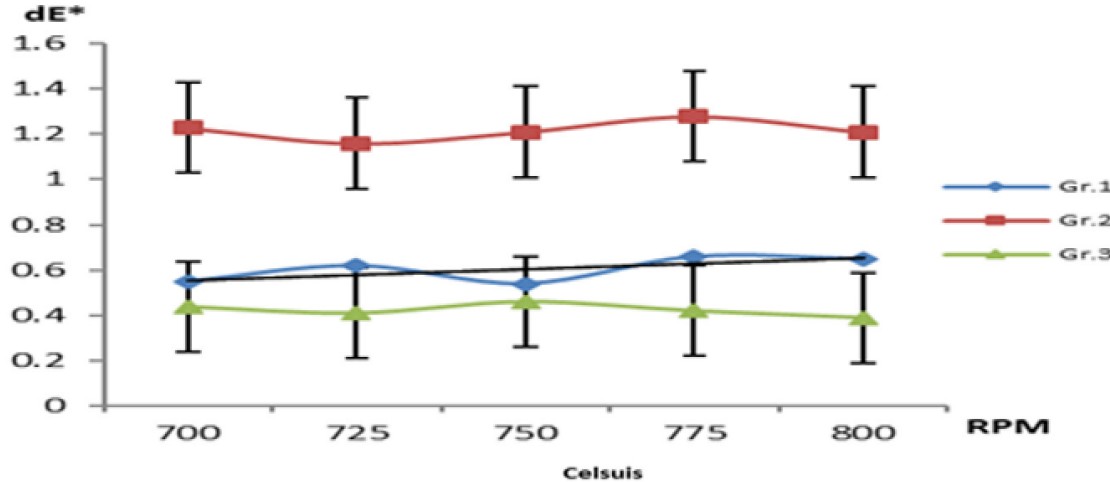

**Figure 3.** Effect of screw speed on dE* for Grade 1, Grade 2 and Grade 3.

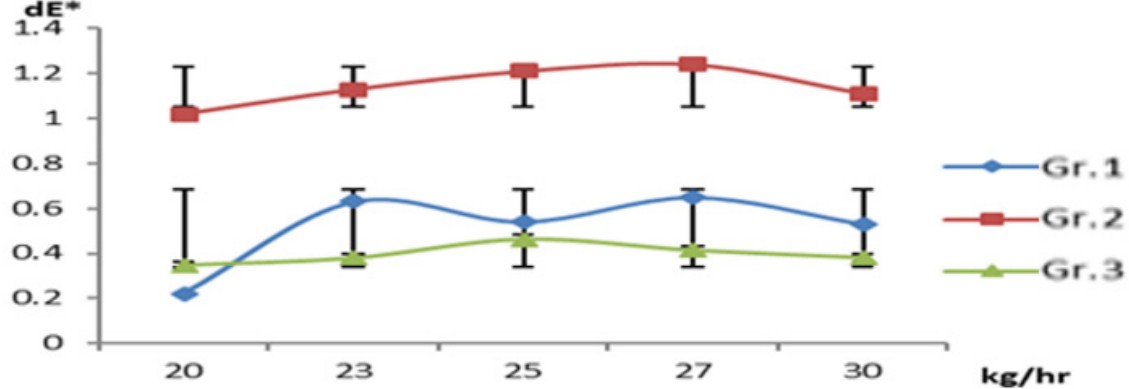

**Figure 4.** The impact of feeding rate on color.

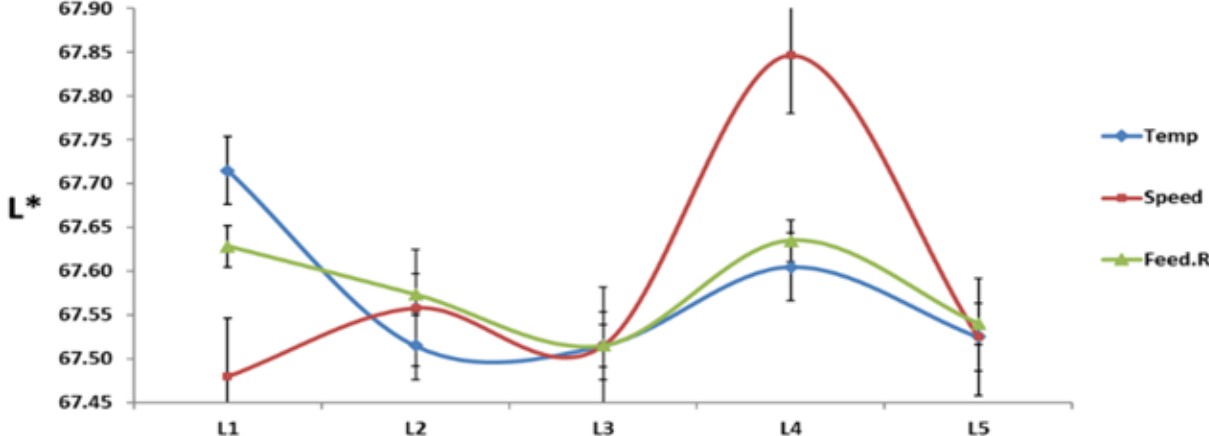

**Figure 5.** Effect of processing parameters on grade 3, L*.

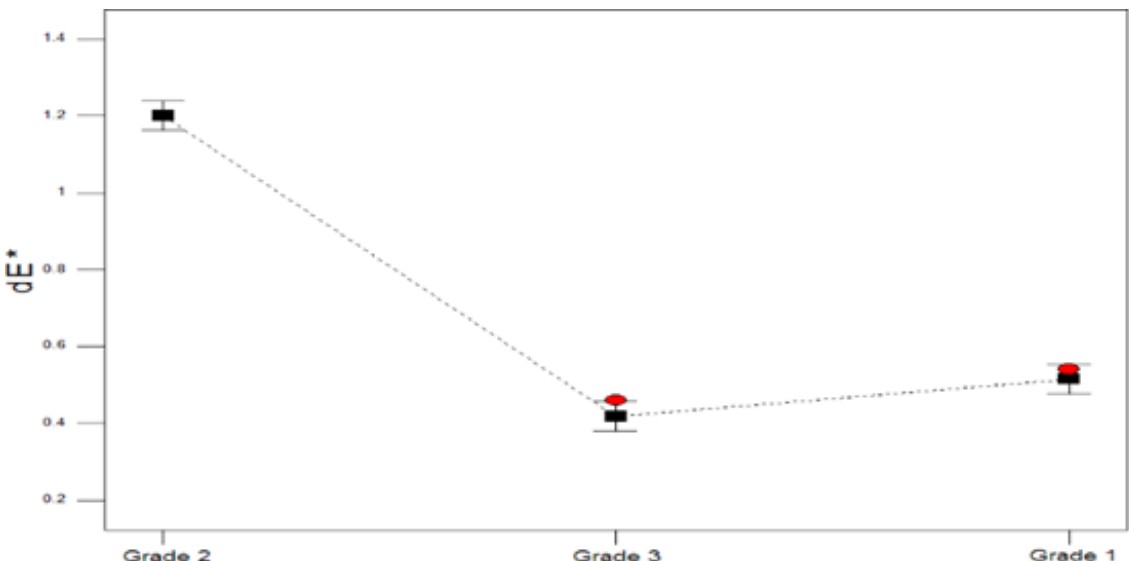

**Figure 6.** Interaction effect of Grade 1, Grade 2 and Grade 3.

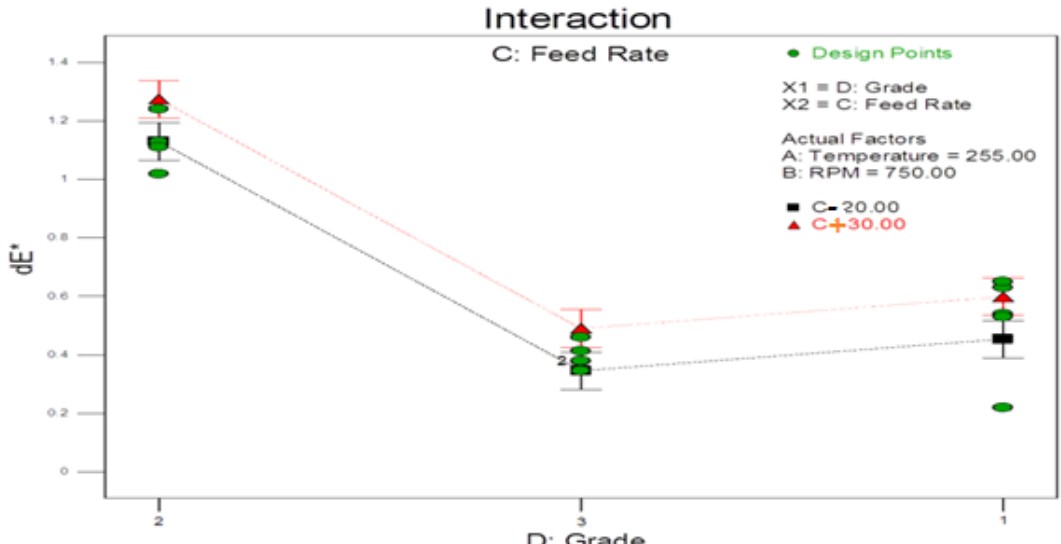

**Figure 7.** Effect of feed rate on dE* for Grade 1, Grade 2 and Grade 3.

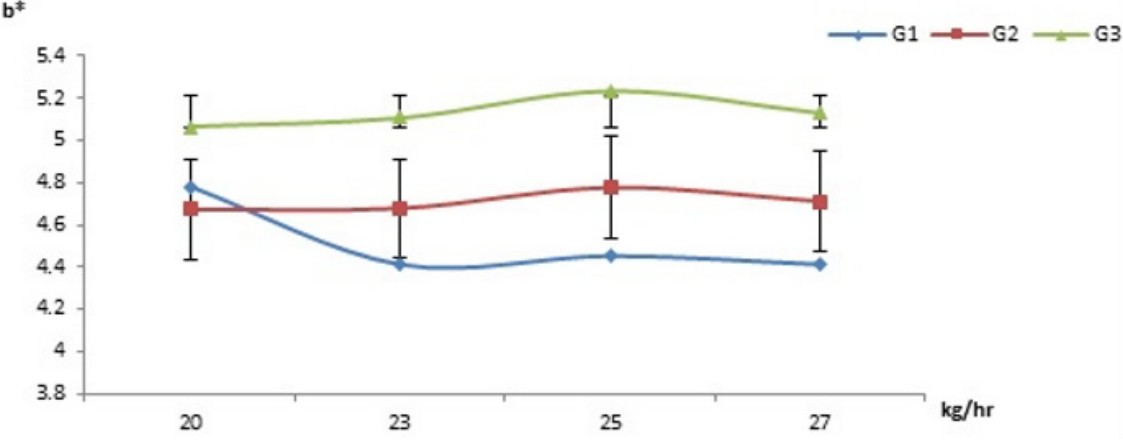

**Figure 8.** Effect of feed rate on b*.

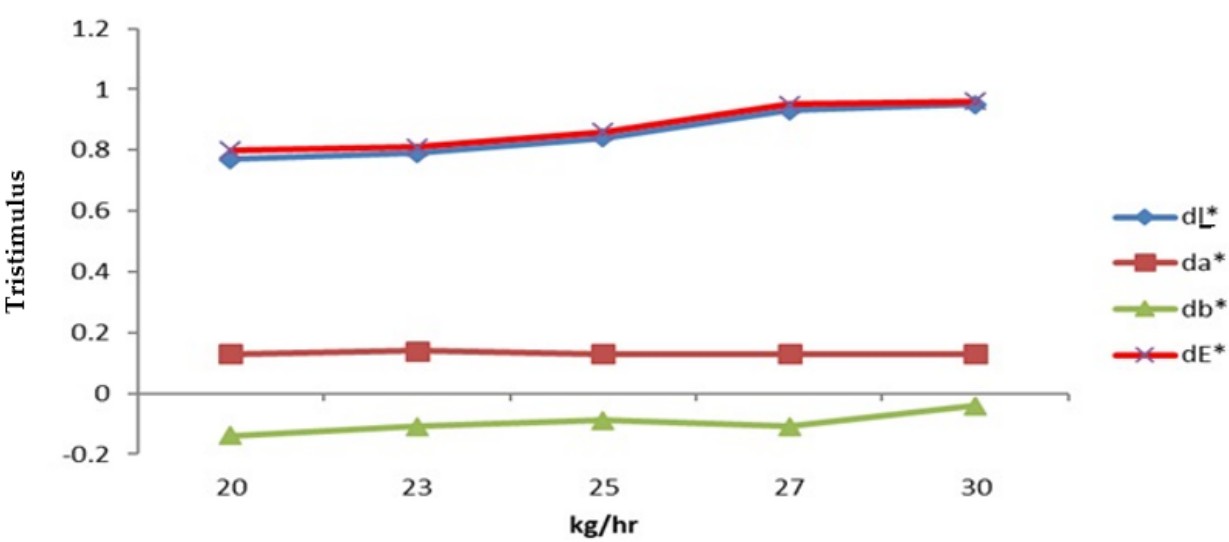

**Figure 9.** Tristimulus versus feed rate on grade 3.

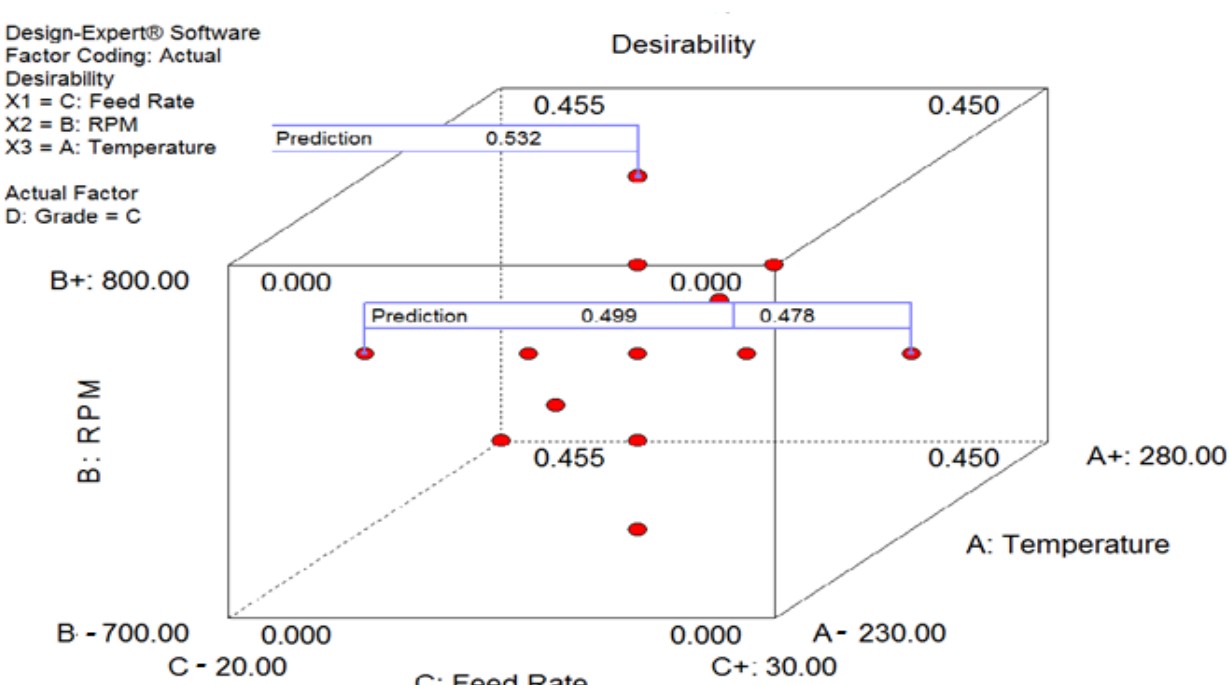

**Figure 10.** Cube desirability between grade 3 and dE*.

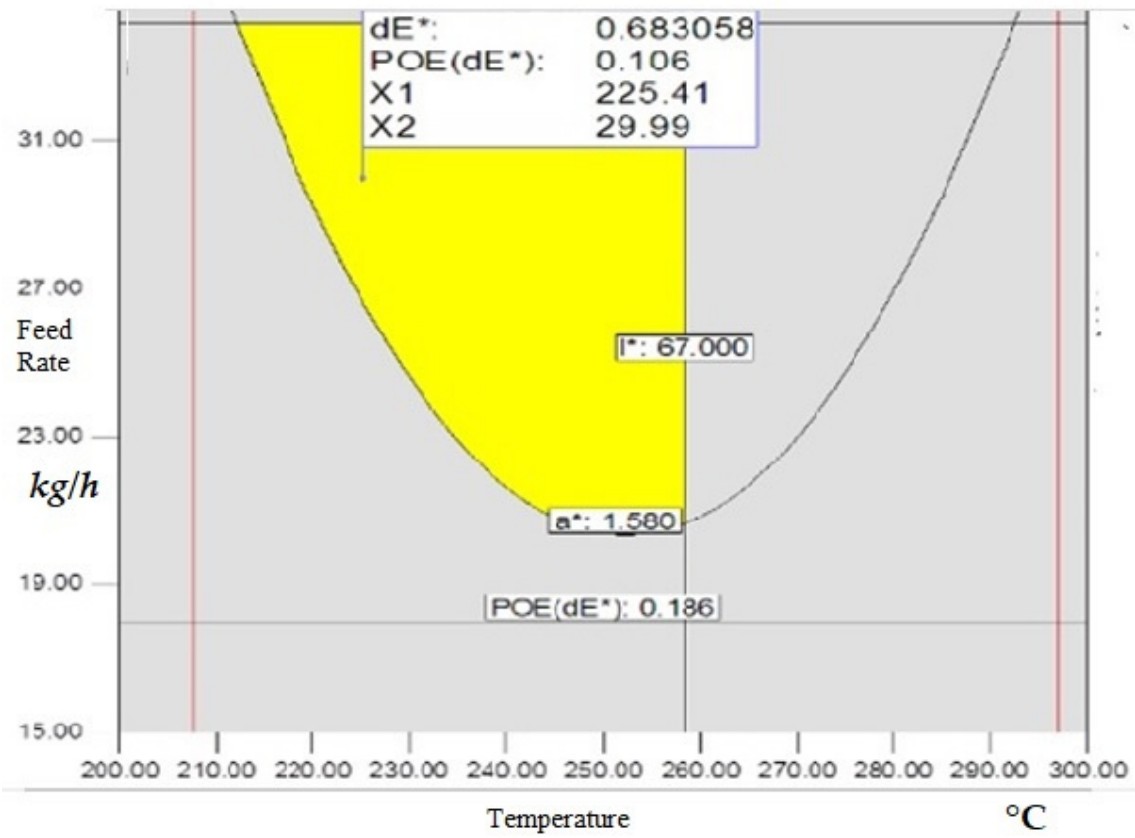

**Figure 11.** Interaction and overlay plot at 750 rpm.

### 3.5. Effect of Temperature on dE*

Figure 2 depicts the fluctuation in dE* about temperature, demonstrating that color output deviation increases somewhat for grades 1 and 3. However, at 280 °C (level five), it became more prominent for both classes—because of the high temperature causing some breakdown in the formulation's resin or additives. When comparing grade 2 to the previous grades, the color output divergence increased dramatically.

### 3.6. Screw Speed Effect on dE*

Figure 3 shows that at 775 and 800 rpm, the responses of grades 1, 2, and 3 to screw speed change are very comparable. This appears to be responsive to screw speed changes due to a higher shear rate, which increases pigment particle dispersion in the extruded material. It is also worth noting that for grade 1, level three had the lowest dE* value. For the three grades, color output begins to improve above 775 rpm. Furthermore, the color output divergence increases dramatically when comparing grade 2 to grades 1 and 3. The shear rate has the same effect on grade 2 as it does on grades 1 and 3.

### 3.7. Feed Rate Effect on Color

Similarly, Figure 4 illustrates the fluctuation in dE* as a function of feed rate. Grades 1, 2, and 3 showed a similar trend to temperature and speed fluctuation (Figures 1 and 2). The feed rate varied between 20 and 30 kg/h. for levels one and five, all three grades demonstrated slight variances from the optimum color output. This could be attributed to a rise in the shear rate at both the maximum and lowest feed rates. In the compounding mixer, this improves the flowability and dispersion of pigments. Furthermore, it shows how they may have a lowering of dE* values differ depending on the feed rate. This could be related to enhanced pigment dispersion, as higher feed rates cause stronger shear, which leads to better pigment dispersion [37,38].

### 3.8. Interactions Effect of L* Values

The interactions between L* and the processing parameters are depicted in Figure 5. All other parameters were fixed, while each was adjusted to five different levels. Only the interactions for L* are shown for the sake of brevity. Figure 5 shows that variations in processing circumstances impact L* values almost like they affect dE* values. A similar trend variation was seen (L1, L2, L3, L4, and L5).

However, the L* values for L3 showed the slightest variance and the same value for the three processing parameters for the target color output, especially for grade 3.

Furthermore, Figure 5 depicts the relationship between processing conditions and (L*) in distinct ways.

Increasing the temperature and weight percentage of PC1 with (higher melt flow index) showed a significant effect in lower viscosity value and decreased color matching values (dE$^*$). The formulation and processes effectively controlled the viscosity, and microtomed plastic sections performed characterizing to different thicknesses and temperatures. The optimal number of particles was increased at higher temperatures and thickness [39]. Characterization of polycarbonate formulation at different temperatures was also analyzed. They were rheologically characterized using the rotational rheometer [40].

### 3.9. Effect of Grades on dE*

Figure 6 shows that dE* values for grades 1 and 2 were greater than those for grade 3. This could be because grade 3 used fewer pigments and had better pigment dispersion than grades 1 and 2.

### 3.10. Grades and Feed Rate Interactions

Figure 7 shows the effect of feed rate on color output across the three grades (when screw speed is 750 rpm and temperature is 255 °C). Figure 7 and Table 7 show that feed rate appears to have the most significant impact on dE* for all three grades. This can be seen at both low and high feed rates, and it could be due to better dispersion.

**Table 7.** Optimum processing values for color output.

| Optimum Processing Values for the Three Grades | | | | | | | |
|---|---|---|---|---|---|---|---|
| Feed Rate Parameter at Fixed Temp and Screw Speed (RPM) | | | | Temp at Fixed RPM and kg/h | | Screw Speed at Fixed Temp and Feed Rate | |
| GRADES | Feed Rate | dE* | Feed Rate | dE* | Temp | dE* | Speed | dE* |
| GRADE 1 | 20 | 0.22 | 30 | 0.53 | 270 | 0.4 | 750 | 0.54 |
| GRADE 2 | 20 | 1.08 | 30 | 1.11 | 280 | 1.23 | 725 | 1.16 |
| GRADE 3 | 20 | 0.34 | 30 | 0.38 | 270 | 0.38 | 800 | 0.39 |

In addition, Table 7 confirms the optimum processing readings for the three grades. The optimum color values are for grade 3 and grade 1.

### 3.11. Effect on b* Values

Figure 8 shows that grades 2 and 3 to b* change responses were comparable. In their formulas, both grades had the same amount of pigment. As a result, this emphasizes the need for having the same pigment composition and the precision of minute pigment loading. It is provided to demonstrate how minor modifications in a formulation can result in major color variations, leading to lot rejection. More precise measurements should be adopted when weighing any pigment amount, especially when working with sensitive formulations. An in-depth study and understanding of pigment interactions can improve first-pass color production [41,42].

This paper aimed to evaluate the influence of various processing parameters on the dispersion quality of polycarbonate compounds. In addition, the influences of param-

eters, pigment size distribution, and morphology on the pigment dispersion were also studied [43].

This study reviews the impact of scanning microscopic methods to evaluate the influence of various processing parameters on the dispersion quality of the polycarbonate compound. Experimental data were compared with historical data records [44]. Figure 9 indicates that the feed rate increased the tristimulus color values of dL*, da*, db*, and dE* for grade 3. It indicates the federate has a significant response on color output.

*3.12. Desirability and Overlay Plots*

The following figures show the desirability and overlay plot between processing parameters and grades. Figure 10 is a 3-D view of the predicted desirability for the interaction of processing parameters in terms of dE* = 0.478. It can be created for each optimal discovery.

Figure 11 illustrates the overlay plot between temperature and feed rate, while screw speed was constant at 750 rpm. In the factor space, the graphical optimization showed the area of possible response values. The yellow region represents the area that meets the required target value, while the gray area represents the area that does not. The points represent the optimum L*, a*, b*, and dE* values for the grades under consideration. The optima occur at 225 °C, 750 rpm, and 29 kg/h, achieving the optimum average value of dE* (0.68) for all grades.

**4. Conclusions**

The current study reveals that different grades respond differently to the desired color output under operating conditions. It is also clear that grade 3 had the best color output value. This could be due to the decreased number of pigments utilized in grade 3's material formulation and, hence, better dispersion of these pigments than in grades 1 or 2. As a result, grade 3 had the lowest MFR readings. A high MFR causes a decrease in viscosity, ultimately breaking the bonds and increasing the flowability of the mixing material.

The impacts of processing factors on color outputs of various grades were investigated, and statistical analysis was used to find correlations between the inputs and outputs. Experiments using general trends (G.T.) and response surface methods (RSM) based on the design of experiments were used to determine the optimum of extrusion settings and color values (DOE). Color outputs and optimal processing conditions were predicted with predictive models.

Different grades produced different color outputs under the same or similar operating conditions.

Finally, it is evident that the three grades had various formulae, but they all had the same color. The optimal color output values and grades were chosen based on our simulation.

From the ANOVA, the *F*-value implied the model was significant for dE*. Feed rate seems to have the most significant effect on dE* of the output color for grade 3. A lowering of dE* values was observed at higher and lower feed rates. This may be due to better dispersion; at increased feed rates, the higher flow generates higher shear, which was associated with better dispersion of pigments. This, in turn, improves the mixture's homogeneity and effectively improves the dispersion of pigments and the quality of the output color. In many cases, mixing resins is required to produce the desired outcomes. However, different resins have different flow rates. The addition of one resin may increase the viscosity of the masterbatch and must go through a melting stage for testing and controlling the quality of the incoming material, which has a substantial impact on color matching. Further research will identify the best processing parameters for various grades and color formulas, resulting in significant waste reduction and faster delivery times for small numbers and prototypes.

**Author Contributions:** J.A. and R.I. were in charge of the study's design. J.A., R.I. and I.T. performed the statistical analyses, and all authors contributed to the interpretation of the results. The final document, which was drafted by J.A., R.I. and I.T. All authors have read and agreed to the published version of the manuscript.

**Funding:** This research received no external funding.

**Institutional Review Board Statement:** Not applicable.

**Informed Consent Statement:** Not applicable.

**Data Availability Statement:** Not applicable.

**Conflicts of Interest:** The authors declare no conflict of interest.

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
