# Peer review of "An Integrative Simulation for Mixing Different Polycarbonate Grades with the Same Color: Experimental Analysis and Evaluations"

_crystals, doi:10.3390/cryst12030423_

Round 1

Reviewer 1 Report

Dear all,

Greetings

Please find enclosed our comments regarding this work

Referenced as: crystals-1606752

Titled as: An Integrative Simulation for Mixing Different Grades Having The Same Color: Experimental Analysis and Evaluations

In my opinion, your paper can be accepted for publication in Crystals Journal after Minor Revisions and after fixing these points:

1) Title: ok

2) Abstract: please add the meaning of CIE (L*, a*, b*and dE*), add the best condition found in your case to get the best color

3) Keywords: just RSD instead of RSD (45 runs)

4) Introduction: add references in your 20 first lines, near to each properties add the appropriate references

5) Tables 2 , 3 and 4 were splitted between two pages

6) 3.5. Effect of Temperature on dE* should not be alone in the bottom of the page

7) pigment dispersion [19]. instead of pigment dispersion. [19]

8) Figure 11 not clear

9) Conclusion: ok which perspectives on your studies

10) References: please updated them 2022 and fellow the journal template

With regards

Author Response

Greetings Reviewer

We appreciate you taking the time to review my work and provide helpful feedback.

Your helpful and insightful feedback influenced the current version's potential enhancements.

I carefully evaluated the remarks and attempted to address them as best I could.

We trust that after rigorous modifications, the manuscript will match your high requirements.

If you have any more constructive suggestions, please let us know.

The point-by-point responses are listed below.

In the manuscript, all changes have been corrected

Sincerely,

Jamal Alsadi, PhD, Ontario Tech University, Canada, jamal.alsadi@ontariotechu.net

Please find enclosed our comments regarding this work

Referenced as: crystals-1606752

  • Title: ok

An Integrative Simulation for Mixing Different Polycarbonate Grades Having The Same Color: Experimental Analysis and Evaluations

2) Abstract: please add the meaning of CIE (L*, a*, b*and dE*)add the best condition found in your case to get the best color

A spectrophotometer is used to determine the color of a compounded plastic batch, which measures three numbers indicating the tristimulus values (CIE L*a*b*).

The lightness axis, which ranges from 0 (black) to 100 (white), is known as the L*-axis (white).

Redness-greenness and yellowness-blueness are represented by the other two coordinates, a* and b*, respectively.

The color difference deviation (Delta E*) from a target is dimensionless.

When dE* approaches zero, however, the greatest favorable color difference value occurs.

3) Keywords: just RSD instead of RSD (45 runs)

Omitted the Number 45

4) Introduction: add references in your 20 first lines, near to each property add the appropriate references

I have added references in the  first 20 lines and all the article

I have additional 24 references(In total 46 references )

5) Tables 2, 3, and 4 were split between two pages

corrected

6) 3.5. Effect of Temperature on dE* should not be alone at the bottom of the page

corrected

7) pigment dispersion [19]. Instead of pigment dispersion. [19]

Corrected

8) Figure 11 is not clear

Corrected-Make it more clear

9) Conclusion: ok which perspectives on your studies

The following lines were added to the conclusions after modifications

The impacts of processing factors on color outputs of various grades were investigated, and statistical analysis was used to find correlations between the inputs and outputs.

Experiments using general trends (GT) and response surface methods (RSM) based on the design of experiments were used to determine the optimum of extrusion settings and color values (DOE).

Predictive models were used to forecast color outputs and optimal processing conditions..

Under the same or similar operating conditions, different grades produced distinct color outputs.

Finally, it's evident that the three grades have various formulae, but they all have the same color. Based on our simulation, the optimal color output values and grade were chosen.

10) References: please updated them in 2022 and follow the journal template

This topic is one of my domain of interest field and I have published many articles 

I have added references in the  first 20 lines and  as well in other parts as required in this  article

I have added around 12-16 references  as  recently published

I have added an  additional 24 references to the previous version (22 ref) . (In total 46 references )

With my best regards

Sincerely,

Dr Jamal Alsadi

Reviewer 2 Report

The authors have presented interesting results in processing. However, a revision addressing the following is required.

  1. In the title, grades of what? Please, specify the material in the title. Based on the title, readers are not able to catch if it is about polymer processing or not.
  2.  There are some standards on the color of polymer such as  ISO 105-A02  and  ISO/CIE 11664-6. The study is about an engineering process, so the methodology should consider the standards.
  3. The manuscript does not treat details of the TSE. Especially, how the machine is setup and how the machine is operated are not described.
  4. There are several studies on the colors of polymer surfaces. The preceding studies are not properly referenced and reviewed. 
  5. It is quite difficult to agree to requirement of the regressions, especially the nonlinear ones. What is the purpose and implications of the regression model? 

Author Response

crystals-1606752

Titled as: An Integrative Simulation for Mixing Different Polycarbonate Grades Having The Same Color: Experimental Analysis and Evaluations

Greetings Reviewer

We appreciate the editors' and reviewers' effort and thoughtful remarks on our submission.I carefully evaluated the remarks and attempted to address them as best I could.We trust that after rigorous modifications, the manuscript will match your high requirements.

If you have any more constructive suggestions, please let us know.

The point-by-point responses are listed below.

In the manuscript, all changes have been corrected (highlighted yellow).We eagerly await the outcome of your outcome evaluation. However, a revision addressing the following  as pin points

Sincerely,

Jamal Alsadi, PhD, Ontario Tech University, Canada, jamal.alsadi@ontariotechu.net

Please find enclosed our comments regarding this work

Referenced as: crystals-1606752

  1. In the title, grades of what? Please, specify the material in the title. Based on the title, readers are not able to catch if it is about polymer processing or not.

Polycarbonate –done

Titled as: An Integrative Simulation for Mixing Different Polycarbonate Grades Having The Same Color: Experimental Analysis and Evaluations

  1.  There are some standards on the color of a polymer such as  ISO 105-A02  and  ISO/CIE 11664-The study is about an engineering process, so the methodology should consider the standards.

We used the methodology to objectively measure a color by developing scales for hue, brightness, and chroma. The L*a*b* color space, also known as CIELAB, is the most prevalent way for quantifying a color.L* stands for lightness, and a* and b* are the chromaticity coordinates in this scheme.A spectrophotometer is used to determine the color of a compounded plastic batch, which measures three numbers indicating the tristimulus values (CIE L*a*b*). This measurement should correspond to the color target values established by quality standards or by the customer.

When compared to the target color, however, measured color values may show variances in terms of dE*, which represents the spatial separation of the output color compared to the target color in the CIE L*a*b* color space.If dE* exceeds a permissible limit, the entire lot produced may be rejected, resulting in waste and delivery time delays.

The effects of processing parameters on the color mismatch of various polycarbonate grades were investigated, and experiments were designed using general trends (GT) and response surface methods (RSM) based on the design of experiments to identify the optimum of extrusion settings and color values.     

  1. The manuscript does not treat details of the TSE. Especially, how the machine is set up and how the machine is operated are not described.

I have added an additional information about the TSE (setup and operations)and more were recorded in the manuscript

Coperion Germany produced the Twin Screw Extruder, which was an intermeshing co-rotating twin-screw extruder (TSE).It had a 27kW motor, a 25.5 mm screw diameter, and a 37 L-to-D ratio. It also included nine barrel heating zones and one die heating zone.

The extrudate was quenched in cold water after departing the die, dried with air, and then turned into pellets using a pelletizer. Temperature, speed, and flow rate were the control process parameters. The combined material was extruded as strands, which were then cooled in water, dried by air, and pelletized.These pellets were then formed into rectangular color chips (3x2x0.1" in size) using an injection molding machine.

            The injection pressure was kept around 28 MPa (1000 PSI) while the temperature was kept around 280°C.The specimen was then dried at lab room temperature before being subjected to more optical microscopic examinations and characterization measurements.

Color measurements were taken with a spectrophotometer at three separate points in each specimen (coupon) to produce the tristimulus values (L*, a*, b*). L*, a*, and b* target values were set to CIE L*, a*, and b* values of L*=67.57, a*=1.43, and b*=4.8, respectively. The color differences were then calculated using the following formulas: dL*, da*, db*, dE*.

To assess the effects of processing parameters such as temperature, feed rate, and screw speed, three polycarbonate grades were compounded under various conditions.

Experiments were run to track each parameter's general trend (GT) while keeping the other values constant.

The best processing parameters for reliable color matching were discovered.The impacts of processing parameters on color formulations for three grades of the same color were investigated.

The analysis of variance was done using Stat-Ease Design-Expert® software (ANOVA).

The impacts of processing factors on color outputs of various grades were investigated, and statistical analysis was used to find correlations between the inputs and outputs.

Experiments to determine general trends (GT) and response surface methods (RSM) based on experiment design were among them (DOE).DOE uses three-level complete factorial designs.

Temperature, speed, and feed rate were all adjusted individually at three different levels while the rest of the parameters remained constant.

The term "general tendencies" was used to describe this (GT).

Temperatures of 230°C, 255°C, and 280°C were chosen, with a speed and flow rate set to the middle values (750 rpm and 25 kg/hr, respectively).

For both speed and flow rate, a similar technique was used.

The selected rotational speeds were 700, 750, and 800 rpm, with flow rates of 20, 25, and 30 kg/hr.

Table 1 shows the experimental design level based on the three process parameters of temperature, feed rate, and screw speed rpm.

The effect of variables on color for three grades was investigated using 45 distinct treatments for three levels of three factors (full factorial response technique). Table 3. Summary of the Design data. Also the below table .1 s included in the table 3

                                              Table1. Setting up a three-level full factorial design

Parameters

Units

3 Levels

L

M

H

Temperature

oC

230

255

280

Screw Speed

rpm

700

750

800

Feed rate

kg/h

20

25

30

The desired color output was L*=67.57, a*=1.43, and b*=4.8 in CIE L*, a*, and b* values.

In order to examine and compare the influence of processing parameters, the statistical analysis of data was performed using the Design-Expert® Software (Version 8, Stat-Ease Inc. USA).The analysis of variance (ANOVA) was used to help identify significant parameters and any interactions between them.

The final goal was to create an equation that could predict L*, a*, and b* tristimulus values while also optimizing the process parameter.To identify interactions and alter the processing settings for color attributes, an ANOVA was utilized.

  1. There are several studies on the colors of polymer surfaces. The preceding studies are not properly referenced and reviewed. 

This topic is one of my domain of interest field and I have published many articles 

I have reviewed and re added around 12-16 references as recently published

I have added an  additional 24 references to the previous version (22 ref) . (In total 46 references )

The mostly topics about  colors of polymers, and properly referenced and reviewed as required

  1. It is quite difficult to agree to the requirement of the regressions, especially the nonlinear ones. What are the purpose and implications of the regression model? 

Regression analysis is a statistical process of fitting continuous functions to a set of independent data points. The goal of regression analysis is to forecast a result based on past data. The purpose of linear and nonlinear regression is to tweak the model's parameters to find the line or curve that best fits our data.

Nonlinear regression is used by scientists to fit a model to their data to determine the best-fit parameter values or to compare the fits of several models. If linear regression fails to provide an appropriate fit, you may need to resort to nonlinear regression. Linear regression is easy to use and interpret, and it provides you with more statistics to assist you to evaluate the model. While linear regression can model curves, nonlinear regression can fit a wider range of curves. However, finding the optimal fit and interpreting the function of the independent variables can take more time.

Multiple linear regression analysis was used to establish the expected response for the L*, a*, and b* functions based on the findings of the ANOVA study, as shown in Table 5. The regression model's parameters and interactions have a significant impact on the projected tristimulus values.

The polynomial equations indicate the quantitative effect of process variables (temperature, speed, and feed rate) on the responses, as well as their interactions. The magnitude of the effect of the respective variables on the replies is correlated with the coefficient values.

Interaction terms and quadratic relationships are represented by coefficients in terms with more than one element or higher order. A positive number suggests an effect that favors optimization, while a negative value shows an effect that is antagonistic to optimization.

With my best regards

Sincerely,

Dr. Jamal Alsadi